# Electrical Manipulation of Spin-Dependent Anisotropy of a Dirac Cone in a Graphene Superlattice with Alternating Periodic Electrostatic and Exchange Fields

**Pattana Somroob [1]** and **Watchara Liewrian [1,2,3,\*]**

1. Theoretical and Computational Physics Group, Department of Physics, Faculty of Science, King Mongkut's University of Technology Thonburi (KMUTT), Bangkok 10140, Thailand
2. Theoretical and Computational Science Center (TaCS), Science Laboratory Building, Faculty of Science, King Mongkut's University of Technology Thonburi (KMUTT), Bangkok 10140, Thailand
3. Thailand Center of Excellence in Physics, Ministry of Higher Education, Science, Research and Innovation, 328 Si Ayutthaya Road, Bangkok 10400, Thailand
* Correspondence: watchara.liewrian@mail.kmutt.ac.th

**Abstract:** We studied the spin-dependent behavior of the electronic properties of alternating periodic potentials applied to finite and infinite graphene superlattices coupled with tunable electrostatic and exchange fields. The band structures were evaluated using the transfer matrix approach. The results of tuning the coupled electrostatic potential and exchange field showed that the spin-dependent anisotropy of a Dirac cone depends on the difference between the amplitude of periodically modulated coupling. Spin-dependent collimation occurs when the modulations become zero-average potentials with the ratio of both periodically modulated strengths equals one, in which one spin can be moved freely, but the other one is highly collimated. In addition, we find that the number of extra Dirac points in the infinite superlattice is spin-dependent. In terms of spin-ups, their number increases with an increase in the strength of both modulated fields. To ensure this calculation, we also compute the conductance of finite periodic modulation at zero energy. It is shown that the peaks of the conductance occur when the extra Dirac point emerges. This result may be utilized to design graphene-based devices with highly spin-polarized collimators.

**Keywords:** graphene; superlattice; band structure engineering; spintronics



## 1. Introduction

The unique properties of graphene derived from its honeycomb lattice, a one-atom-thick carbon material, have received a great deal of attention in both the technological and fundamental paradigms over the past decade. One of graphene's most remarkable properties is its electronic band structure. At low energy, the charge carriers in graphene can be approximated as relativistic quasiparticles governed by the effective Dirac equation, where their Fermi velocity and sublattices in graphene play the same roles as the speed of light, $v_0 \approx 10^6$ m/s, and pseudospin, respectively [1,2]. As a result, several relativistic quantum phenomena, such as Klein tunneling [3] and the room-temperature quantum Hall effect [4], can be observed in graphene.

Spintronics, referring to devices that can use the spin quantum number, is used in a variety of technological applications, including data storage devices, magnetic sensors, and quantum information-processing devices. Due to its weak spin–orbit interaction, long-distance diffusion of spin transport, excellent spin injection properties, and the longest spin-relaxation length at room temperature, graphene is capable of hosting the spintronics characteristic [5**?** ]. It was recently reported that graphene can enable spintronics via the magnetic proximity effect. The interactions between the interface of the heterostructures of graphene and ferromagnets, such as europium chalcogenides (EuS and EuO) [7–10], or

ferrimagnets, such as yttrium ion garnet (YIG) and cobalt ferrite [11], have been proposed to generate the proximity effect. The proximity effect produces an effective exchange field that causes spin splitting in the band structure. An adjacent magnetic insulator of EuS has also been experimentally reported to induce the proximity effect locally in graphene [12], which sheds light on the modulation of spintronic features in graphene.

Meanwhile, the study of graphene superlattices, which are periodic patterns overlaid on a graphene lattice, remains an active area of research for the purpose of modifying electronic properties and exploring new states of matter. In recent years, various intriguing phenomena have been discovered through the use of various superlattice modulations, such as the emergence of superconducting, strange metal phases, and skew scattering of chiral electrons [13–15]. One practical approach to utilizing superlattice modulation in graphene is through 1D periodic modulation. Research has shown that the band structure of graphene can be significantly altered, depending on the type and pattern of modulation [16]. For example, the miniband of the superlattice becomes anisotropic in shape for 1D superlattices of periodic electrostatic potentials (PEPs) [17], which can lead to electron-optic supercollimation in graphene [18–20]. Moreover, graphene superlattices can exhibit multiple Dirac points in addition to the original ones, called extra Dirac points [21–28]. These extra Dirac points tend to be more robust against lattice disorder compared with the original Dirac point [29]. The use of a periodic magnetic field can also lead to a reduction in the group velocity of graphene, resulting in extra finite-energy Dirac points [30–32]. Furthermore, modulation of the periodic exchange potential (PExP) has been suggested as a method to achieve perfect spin polarization and wave vector-based spin filtering in graphene [33–35]. Although several studies have been proposed on graphene superlattices, there have been no reports on the possibility of controlling the spin-dependent anisotropy of the Dirac cone in modulated magnetic graphene superlattices.

In this work, motivated by the preceding literature, we investigate the electronic structure of the alternatively aligned PEP/PExP superlattice pattern that PEPs alternate, while zero-averaged, along the periodic modulation. This superlattice design could be used to control spin dependence. This study aims to reveal interesting applications of graphene band structure engineering for nanospintronic devices based on Dirac materials.

This paper is structured as follows. First, using a transfer matrix approach, we investigate the spin-dependent anisotropic behavior of the miniband of the superlattice of PExP coupled with PEP by visualizing its Fermi contour and computing its group velocity in the low-energy regime. For the sake of simplicity, we assume that both modulated fields are zero-averaged. Second, we calculate the spin-dependent extra Dirac points at zero energy and the corresponding conductance of a finite periodic modulation resonance at zero energy. Third, we consider the spin-splitting effect on the zero-average periodic field. Finally, in the conclusion section, we summarize our findings.

## 2. Theoretical Framework

In this report, the electronic properties of graphene modulated by a one-dimensional superlattice of PEP and PExP are considered. As shown in Figure 1, the graphene sheet is modulated by a succession of electrostatic gates and a magnetic insulator stripe. To apply the proximity of the exchange field on graphene, as reported in [36], graphene is exfoliated on a YIG substrate. The electrostatic gate pattern can be modeled as a periodic function $U(x)$ aligned with graphene along the direction $x$, where its total fields along this direction are zero. On the other hand, the magnetic insulator stripes can be modeled as effective exchange fields $M(x)$ that are coupled to $U(x)$ in the same direction. For simplicity, these periodic modulations are approximated as the Kronig–Penney model superlattice, which are simplified as multiple rectangular barriers. For each supercell, the profile of these external fields is divided into two different regions that are labeled as $A$ and $B$, where the external electrostatic (and the exchange fields) in region $A$ (and $B$) are assigned as $U(x) = U_A$ (and $U_B$) and $M(x) = M_A$ (and $M_B$). These can be defined as $x$ $[(n-1)L, w_a + (n-1)L]$ for $U(x)$ (and $([w_a + (n-1)L, nL]$ for $M(x)$), where $L$ is the

superlattice constant, $w_a$ is the width of region, $A$ and $n$ indicates the number of periodic barriers ($n = 1, 2, 3 \ldots$).

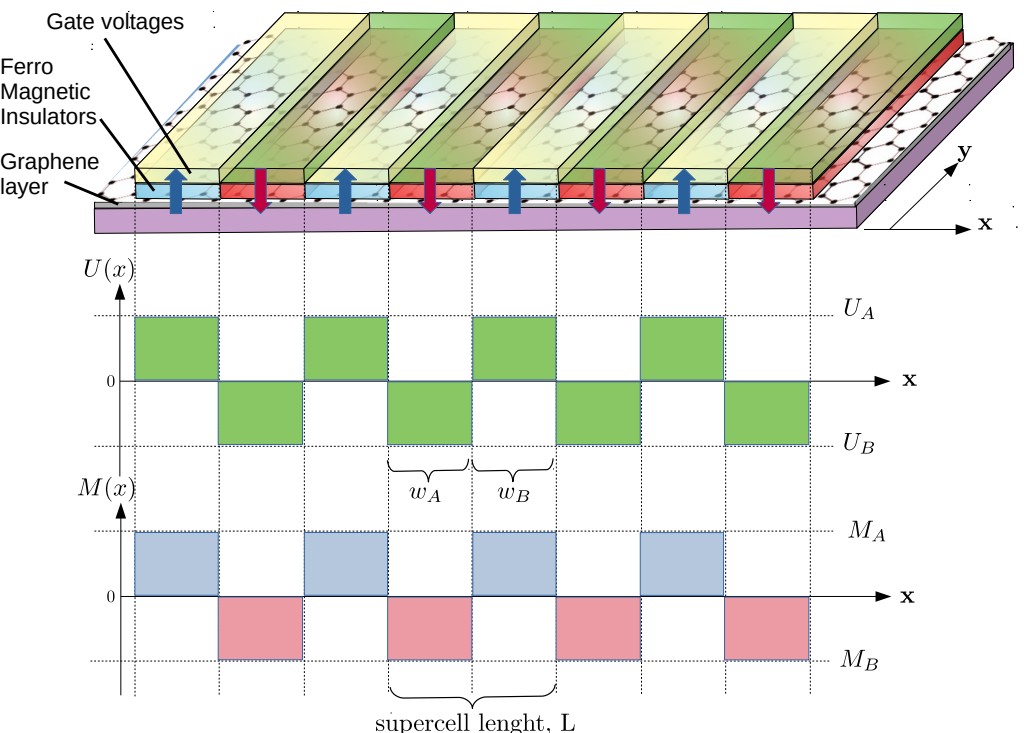

**Figure 1.** Schematic illustration of PEP-PExP graphene superlattice produced by a series of magnetic insulator and electrostatic gates. Here, $L$ is the superlattice constant separated into two regions, labeled by $A$ and $B$, where their widths are $w_A$ and $w_B$, respectively. This periodic field profile is approximated as a series of square barriers, in which each part contains the exchange field and the electrostatic with barrier heights of $u_A(u_B)$ and $m_A$ ($m_B$) for region $A$ ($B$), respectively.

In this paper, we reduce this problem to the low-energy limit, and thus the electronic structure of our model can obey the effective Dirac Hamiltonian:

$$\hat{H}(x,y) = -i\hbar v_0(\tau_z\sigma_x\partial_x + \sigma_y\partial_y) + s_i M(x) + IU(x) \tag{1}$$

where $v_0 \approx 10^6$ m/s is the Fermi velocity, $\sigma_x$ and $\sigma_y$ are the Pauli matrices on the sublattice space $(x, y)$, respectively, and $s_i$ is the vectorial spin matrix that acts on the electronic space $(k_x, k_y)$. Note that Equation (1) is a $4 \times 4$ operator due to the effective exchange field term. For the sake of simplicity, in this paper, we reduce this problem to a $2 \times 2$ problem by setting $s_i$ to be only in the $z$ direction. Thus, Equation (1) can be decoupled into two $2 \times 2$ Hamiltonians:

$$\hat{H}_\xi(x,y) = -i\hbar v_0(\sigma_x\partial_x + \sigma_y\partial_y) + \xi M(x) + IU(x) \tag{2}$$

where $\xi$ is the spin index ($\xi = 1$ for spin up and $\xi = -1$ for spin down). Therefore, the Dirac equation reads as follows:

$$\hat{H}_\xi(x,y)\Psi_\xi(x,y) = E\Psi_\xi(x,y), \tag{3}$$

where $\Psi_\xi(x,y)$ is the spinor wave function of an electron with a spin index $\xi$. Additionally, we presume that the size of the graphene sheet in the y direction is very large. Accordingly, the effect of the edge boundary can be neglected due to the translation symmetry. Consequently, the wave function of Equation (3) reads $\Psi_\xi(x) = \psi_\xi(x)e^{ik_y x}$, where $\psi_\xi$ is a

two-component spinor for an electron with a spin index $\xi$. Here, we reform Equation (3) to be a linear differential equation that reads

$$i\frac{\partial_x \psi_\xi(x)}{\partial x} = \hat{h}_\xi(x)\psi_\xi(x) \tag{4}$$

where $\hat{h}$ is the linear operator:

$$\hat{h}_\xi(x) = \begin{pmatrix} ik_y & \frac{U(x)+\xi M(x)-E}{\hbar v_0} \\ \frac{U(x)+\xi M(x)-E}{\hbar v_0} & -ik_y \end{pmatrix}. \tag{5}$$

The general solution to Equation (4) is

$$\psi_\xi(x) = \hat{\mathcal{P}}_x \exp\left(-i\int_{x_0}^x \hat{h}_\xi(x_1)dx_1\right)\psi_\xi(x_0), \tag{6}$$

where $\hat{\mathcal{P}}_x$ is the spatial ordering operator [37]. In our model, $U(x)$ and $M(x)$ in $\hat{h}_\xi(x)$ depend on $x$ as mentioned above, and $\hat{h}_\xi(x)$, corresponding to regions $A$ and $B$, becomes $\hat{h}_A(\xi)$ and $\hat{h}_B(\xi)$, respectively. In each region, this operator is constant, and they communicate with each other. Subsequently, Equation (6) can be reduced to

$$\psi_\xi(x) = \hat{\mathcal{M}}\psi(x_0) = \exp\left(-i[x-x_0]\hat{h}(\xi)\right). \tag{7}$$

For the $j$ cell in the modulation cell, we have

$$\mathcal{M}_j(\xi) = \begin{pmatrix} \frac{\cos(q_j\Delta x - \theta_j)}{\cos\theta_j} & \frac{\sin(q_j\Delta x)}{\cos\theta_j} \\ \frac{\sin(q_j\Delta x)}{\cos\theta_j} & \frac{\cos(q_j\Delta x + \theta_j)}{\cos\theta_j} \end{pmatrix}, \tag{8}$$

where $q_j$ is the wave vector for the $j^{\text{th}}$ barrier, $\theta_j$ is the injection angle for each $q_j$, and $\Delta x$ is the spatial width of each cell.

In this work, we separated our calculation of the electronic properties of graphene modulated with PExP and PEP into infinite and finite superlattice approach models. For an infinite superlattice, we estimated the electronic behavior by setting $n \to \infty$. In this case, we could then utilize Bloch's theorem to assign the periodic condition $\psi(L) = \exp i\kappa_x L$. (Here, $\kappa_x$ is the Bloch wave vector of the 1D superlattice.) Consequently, with Equation (7), the dispersion relation becomes $2\cos(\kappa_x L) = \text{Tr}(\hat{\mathcal{M}}_{AB})$. For simplicity, we introduced dimensionless units of energy $\varepsilon_0 = \hbar v_0/L$. Therefore, $E \to \varepsilon\varepsilon_0, U(x) \to u(x)\varepsilon_0$ and $M(x) \to m(x)\varepsilon_0$. Thus, the dispersion of this superlattice in dimensionless formulation is

$$\cos(\kappa_x L) = \cos(q_A a)\cos(q_B(L-a)) + \frac{Q}{q_A q_B}\sin(q_A a)\sin(q_B(L-a)) \tag{9}$$

where $q_A = \sqrt{-\varepsilon + \xi m_A + u_A - k_x^2}$, $q_B = \sqrt{\varepsilon + \xi m_B + u_B - k_x^2}$ and $Q = -\varepsilon^2 + (\xi m + u)^2 + k_x^2$.

For a finite superlattice, we assume that the wave function propagates from region $A$ to region $B$ with $n$ periods. Therefore, Equation (7) becomes

$$\psi_\xi(NL) = \prod_{n=1}^N \hat{\mathcal{M}}_B \cdot \hat{\mathcal{M}}_A \psi_\xi(0) \tag{10}$$

Here, we rewrite the product term in matrix form:

$$\prod_{n=1}^N \hat{\mathcal{M}}_B \cdot \hat{\mathcal{M}}_A = \begin{pmatrix} X_{11} & X_{12} \\ X_{21} & X_{22} \end{pmatrix}, \tag{11}$$

where $X_{ij}$ is the value of the element in the matrix. According to the transfer matrix method, the transmission coefficient can be numerically calculated as follows:

$$t_\xi = \frac{2\cos\theta_0}{\left(X_{22}e^{-i\theta_0} + X_{11}e^{i\theta_N}\right) - X_{12}e^{i(\theta_N-\theta_0)} - X_{21}}. \tag{12}$$

For both the infinite and finite model, the conductance along the modulation direction (x-axis) can be calculated in the zero temperature limit using the Landauer–Büttiker formula approach [38]:

$$\sigma = \sigma_0 \int_{-\infty}^{+\infty} dk_y T(k_y, \varepsilon), \tag{13}$$

where $T(k_y, \varepsilon)$ is the transmission probability of $k_y$, while $T(k_y, \varepsilon) = t_\xi^* t_\xi$ and $\sigma_0 = 2e^2\varepsilon/\hbar$.

## 3. Results and Discussion

In this section, we discuss the electronic properties of graphene modulation by 1D PExP and PEP, which are illustrated in Figure 1. The corresponding superlattice was modeled as the Kronig–Penney model with a superlattice constant of $L$. Each supercell was divided into two equal regions labeled $A$ and $B$, which were assigned as $w_A$ and $w_B$, respectively, where $w_A = w_B = L/2$. Here, the corresponding heights of the electrostatic and exchange fields in region $A(B)$ are $u_A(u_B)$ and $m_A(m_B)$. Recently, it has been reported that graphene on $SiO_2/Si$ can have an exceptionally long-distance spin transport capability of 45 μm and a spin diffusion length of 13.6 μm at room temperature [**?** ]. Furthermore, gate-controlled graphene superlattices capable of creating a graphene superlattice with a scale of periodicities as low as 35–150 nm can also be fabricated [40,41]. For these reasons, it is possible to fabricate a multi-periodic modulation of PEP/PExP on graphene that can preserve the spin properties with ballistic transport.

### 3.1. Spin-Dependent Anisotropy Miniband

Let us consider the spin-dependent anisotropic behavior of the superlattice miniband using the group velocity near the zero-energy Dirac point. To simplify the following analysis, we introduce a periodic pattern with a fixed superlattice Dirac point for both spin-up and spin-down at the zero-energy point, which can be divided into two categories: in-phase and out-of-phase combinations, as shown in Figure 2a,b, respectively. In this step, we initialize both PExP and PEP, oscillating between $-\pi$ and $\pi$ in the supercell. For example, $u_A = -u_B$ and $m_A = -m_B$, in which $w_A = w_B = L/2$. Here, we restrict our study to the $k_y$ axis only because spatial modulation strongly affects the $k_y = 0$ axis, and electron propagation along the $k_x$ axis does not change due to Klein tunneling. Then, we numerically calculate Equation (9) to serve such conditions for the in-phase and out-of-phase categories as shown in Figure 2c,d. According to Figure 2c, the spin-up electron is strongly periodically modulated for the in-phase modulation pattern, but the spin-down electron is weak. As we can see, the miniband broadens significantly in the $k_y$ direction due to the combination of electrostatic and exchange interaction. As shown in Figure 2a,b, the summation of PExP and PEP vanishes for spin-down (spin-up) modulation patterns but increases for spin-up modulation patterns. This extremely spin-dependent condition can result in the miniband's highly anisotropic behavior. Furthermore, we believe that this spin-dependent anisotropic behavior can be switched between spin indexes by simply switching between in-phase and out-of-phase configurations.

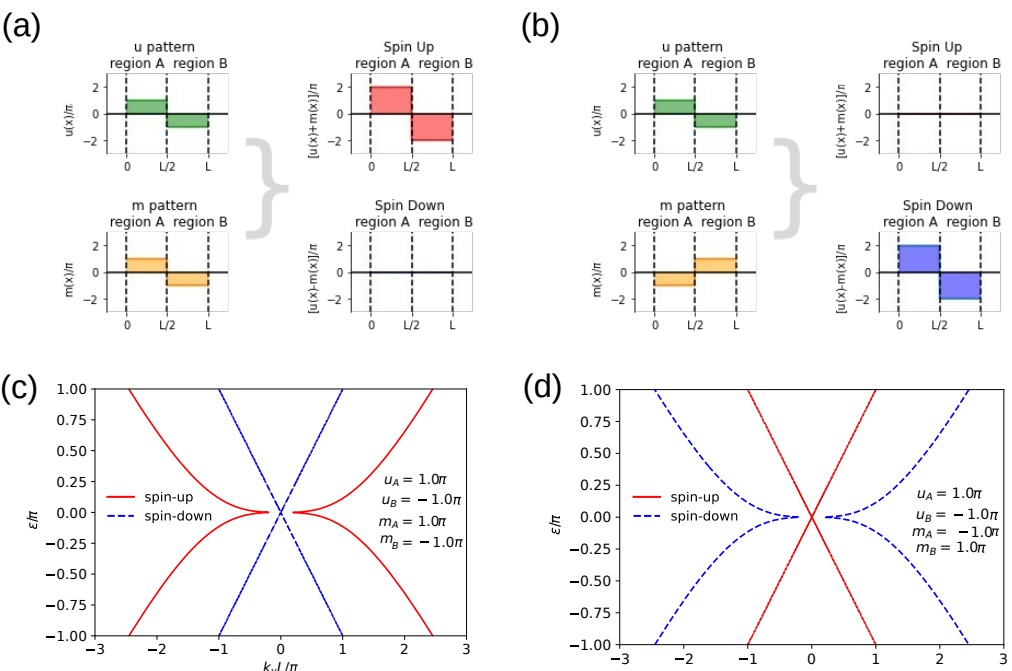

**Figure 2.** Model of PEP-PExP graphene superlattice that can locate original miniband Dirac point at $(k_x, k_y) = (0,0)$. (**a**) In-phase and (**b**) out-of-phase coupling patterns between PEP and PExP, where the PEP pattern of each region is $u_A = -u_B = 0.5\pi$, and the PExP of the in-phase (out-of-phase) pattern is $m_A = -m_B = 1\pi$ ($-m_A = m_B = 1\pi$). The corresponding miniband structures for (**c**) the in-phase and (**d**) out-of-phase patterns are shown for each spin index as a function of $k_y$.

To understand the impact of PEP on the preceding discussion, we only examined the in-phase modulation configuration, which has mirror symmetry with the out-of-phase one. To accomplish this, we set $u_A = -u_B = u$ and $m_A = -m_B = m$, where $u$ and $m$ are positive values. This miniband configuration is evaluated using Equation (9), as depicted in Figure 3a,c, for spin-up and spin-down, respectively. As $m$ increased, the miniband broadened significantly for spin-up but barely deformed for spin-down.

We quantified this spin-dependent reshaping of the miniband by taking its group velocity into account. For the sake of simplicity, we only considered the low-energy regime. As a result, we could approximate Equation (9)'s dispersion relation with the parameter introduced via second-order Taylor expansion regarding the zero-energy point ($\varepsilon = 0$, $k_y = 0$, and $k_x = 0$). Next, we solved this expansion for the dispersion relation to achieve the approximated linear dispersion relation:

$$\varepsilon(\xi) = v_0 \sqrt{k_x^2 + \left[ \frac{4}{(u + \xi m)^2} \sin^2\left( \frac{u + \xi m}{2} \right) \right]^2 k_y^2}, \tag{14}$$

The numerical calculation of Equation (14) is shown in Figure 3b,d for spin-up and spin-down, respectively. Here, the approximated group velocity $v$ is defined as $\partial\varepsilon/\partial k$, where $v = (v_x, v_y)$. As mentioned before, there was no band deformation in the $k_x$ direction. Hence, we could focus on the anisotropic behavior using $k_y$ only. Therefore, the group velocity in the $k_y$ direction can be written as

$$v_y(\xi) = \frac{4}{(u + \xi m)^2} \sin^2\left( \frac{u + \xi m}{2} \right) v_0. \tag{15}$$

We calculated Equation (15) on a contour of $u$ and $m$ for spin-up and spin-down as shown in Figure 4a,b. As $m$ and $u$ grew, the spin-up velocity gradually decreased. The

spin-down velocity, on the other hand, was slightly suppressed when $u \approx m$ (shown in the red area of Figure 4b, except the line $u = m$, which shows that the group velocity was at its maximum), but it was successively suppressed when $u < m$ or $u > m$. Figure 4c depicts a cross-section of these contours. The plot depicts the spin-dependent group velocity behavior $v_y$ as a function of $u$ with a constant value of $m$. According to Equation (15), $v_y$ is highly suppressed when $u + \xi m = 2n\pi$, where $n = 1, 2, 3, \ldots$. On the other hand, $v_y$ can reach the maximum value ($v_y / v_0 = 1$) when $u$ and $m$ satisfy $\sin[(u + \xi m)/2] = (u + \xi m)/2$, which can occur only if $u + \xi m \approx 0$ for instance.

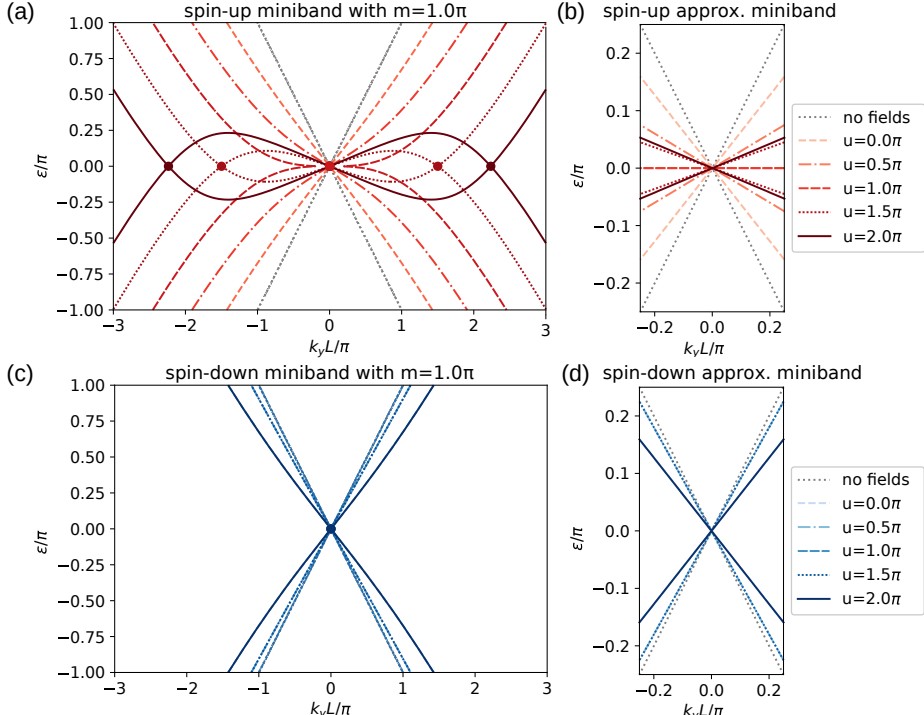

**Figure 3.** Miniband structure of zero-averaged PEP-PExP graphene superlattice as a function of $k_y$ with $m = 1\pi$ when varying magnitude of $u$ for (**a**) spin-up (and (**c**) spin-down) and (**b**) its corresponding approximation with the superlattice Dirac point for (**b**) spin-up (and (**d**) spin-down).

These assumptions show that we can tune $u$ or $m$ in this model to change the anisotropic miniband for each spin. When $u = m = n\pi$, the difference between $v_y$ for spin-up and spin-down is at its maximum, as shown by the black dots in Figure 4d. As a result, under this condition, one of the spin index bands is broadened to zero energy (parallel to the $k_x$ direction), while the other remains unchanged ($v_y \approx v_0$). Compared with the work in [17], PExP was added in our model, which was alternatively aligned to the PEP modulation. Even though they also reported anisotropic behavior with major differences in superlattice patterns, the spin-dependent features were taken into account in our results and could be tuned without spin-splitting despite an exchange field being applied. As reported in [18], the superlattice modulation in graphene can refer to the supercollimation effect, which is an electron beam with a long travel distance, referred to as an optical collimation. According to our findings, this spin-dependent anisotropic behavior can also result in the spin-dependent supercollimation effect.

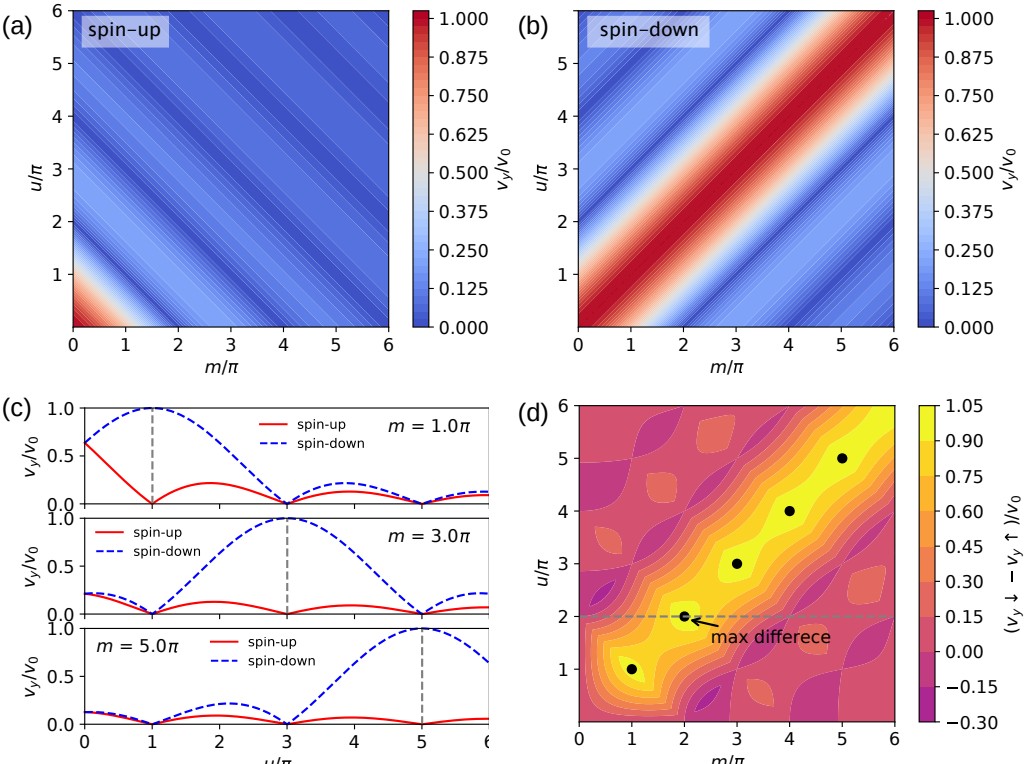

**Figure 4.** Contour plot of group velocity of the electron in zero-averaged PEP-PExP graphene superlattice versus $u$ and $m$ for (**a**) spin-up and (**b**) spin-down. (**c**) The corresponding line plot of the group velocity with varying of $u$ with $m = 1.0\pi$, $3.0\pi$, and $5.0\pi$. (**d**) Contour plot of the group velocity difference between spin-up and spin-down with spots (black dots) that indicate max difference points of the group velocity difference on $(u, m)$.

Figure 5 shows the energy surfaces obtained from Equation (9). The dispersion of this supercell configuration is near the zero-energy point. Depending on the values of $m$ and $u$, the shape of the energy surfaces can take the form of a circle, an ellipse, or a series of straight lines. The approximation of the velocity mentioned above can indicate critical shapes, such as circles and straight lines (Equation (15)). Under $m = u = n\pi$, the energy surfaces of both spin structures differed greatly from each other (see Figure 5a). Because the miniband was critically broadened on the $k_y$ axis, which corresponded to $v_y = 0$, the energy surface exhibited straight lines parallel to the $k_x$ axis during spin-up. On the contrary, for spin-down, circles were observed, as in the pristine graphene, which corresponded to $v_y = v_0$, and there were no restrictions on electron motion. The velocity results of both spin indexes were zero, and the energy surfaces of both spin indexes are shown as straight lines in Figure 4c (see Figure 5c). As a result of this superlattice model, the spin-up electron can beam as electron-optic collimation, but the spin-down electron cannot, and vice versa. This feature can be realized as an electric controllable-spin supercollimator that can regulate each spin-index electron wave. For instance, one spin is highly collimated but not the opposite one.

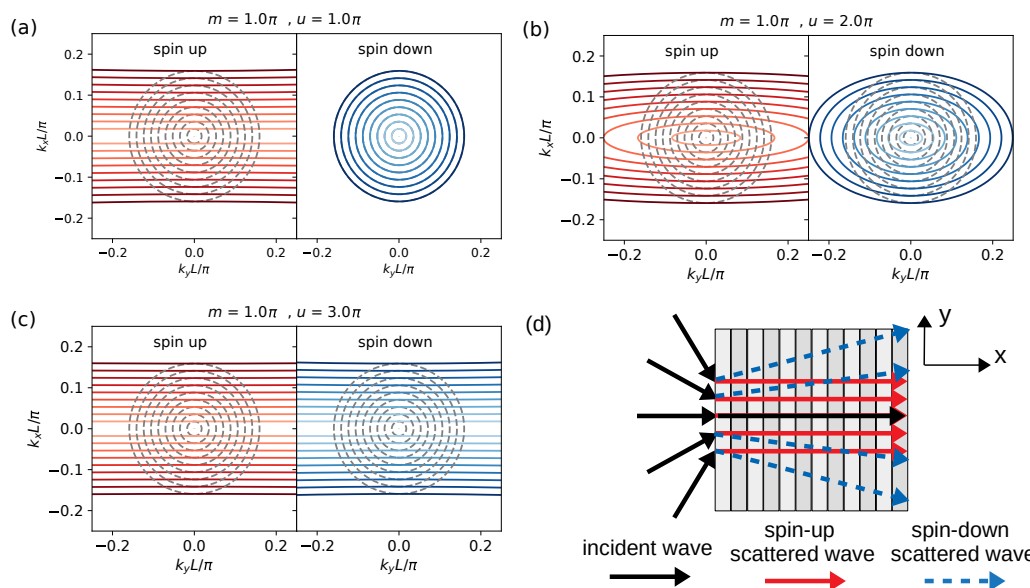

**Figure 5.** Energy contour of PEP-PExP graphene superlattice miniband structure with fixed $m = 1.0\pi$ for spin-up (left) and spin-down (right) for (**a**) $u = 1.0\pi$, (**b**) $u = 2.0\pi$, and (**c**) $u = 3.0\pi$. (**d**) The demonstration of the electron trajectory propagates across the PEP-PExP graphene superlattice of spin-up (red line) and spin-down (dashed blue line).

### 3.2. Spin-Polarized Transport and Extra Dirac Points

According to Figure 3a, when $u$ and $m$ reach certain critical values, the line $u = 1.5\pi$ ($u = 2\pi$) with $m = \pi$ and $\xi = 1$ (spin-up) exposes two additional touching points at zero energy: the so-called extra Dirac points. As previously stated, the number of these extra Dirac points is proportional to the modulation pattern's field strength, which corresponds to the result of the work in [40]. However, each spin index is affected by a different total modulated field. The spin-up miniband experience is more modulated than the spin-down one, and the spin-up experience generates more Dirac points than the opposite one. As a result, the number of extra Dirac points in this model is spin-dependent. To investigate the emergence of extra Dirac points in depth, we first computed the possible locations of these extra Dirac points on the $k_y$ axis by reformulating Equation (9) with $\varepsilon = 0$ and $k_y = 0$. As a result, we obtained $\cos((u + \xi m)^2 - k_y^2) = 0$, which can be rewritten as

$$k_{exDP_N}(\xi) = \pm\sqrt{(u + \xi m)^2 - 4n^2\pi^2},\tag{16}$$

where $n = 1, 2, 3, \ldots$. As shown in Figure 3a, for spin-up, there were two additional Dirac points located at $k_y = \pm 1.50\pi$ for $u = 1.50\pi$ ($k_y = \pm 2.87\pi$ for $u = 2.0\pi$), and for spin-down, there were no such additional points. This mark was asserted by Equation (16), showing that the extra Dirac point locates $k_{exDP_N}$ only if $(u + \xi m)^2 > 4n^2\pi^2$, which is a real value. Then, the number of these extra Dirac points for the spin index $\xi$ can be written as

$$N(\xi) = \frac{u + \xi m}{2\pi}.\tag{17}$$

The contours of Equation (17) vs. $u$ and $m$ for spin-up and spin-down are plotted in Figure 6a,b, respectively.

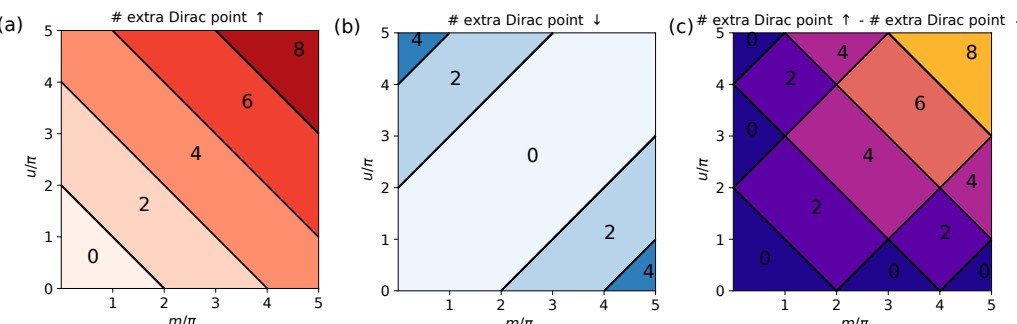

**Figure 6.** Contour plot of extra Dirac point number at zero energy of zero-averaged PEP-PExP graphene superlattice with variations of *u* and *m* for (**a**) spin-up, (**b**) spin-down, and (**c**) different numbers of the extra Dirac points of spin-up and spin-down.

The patterns of extra Dirac points emerging from two spin indexes were completely different. In spin-up (spin-down), the extra Dirac points increased while either *m* or *u* increased (decreased). In particular, the conditions resembled its group velocity reaching critical values $v_y = 0$ in the transition condition between different numbers of extra Dirac points (see Figure 4a,b). The group velocity reached zero when a new pair of extra Dirac points emerged at the original Dirac point. This situation occurred repeatedly during spin-up when $|u + m| \mod 2\pi = 0$. In contrast, for spin-down, $|u - m| \mod 2\pi = 0$. With each recursive emergence, the distance between the original Dirac point and the extra point ($|k_{exDP_N}|$) of the previously emerged extra Dirac point grew. To calculate the conductance in zero mode, we calculated Equation (13) with $\varepsilon = 0$ along the finite periodic modulation direction to identify this phenomenon. The conductance of spin-up and spin-down is plotted as a function of *u* with different *m* and several periodic modulations *N* in Figure 7. There are resonance peaks of conductance at specific values for each fixed *m* plot, and these peaks become sharper as *N* increases. When we compare the spin-up and spin-down curves for each *m*, we can see that their resonance peaks were in different locations. As is known, these peaks occur when the new extra Dirac points emerge at the original Dirac point. When comparing Figure 7a,b, it can be seen that these peaks occurred around the critical value at which new extra Dirac points appeared. As a result, we could use this conductance change to practically ensure the phenomena discussed above, namely the strong collimation.

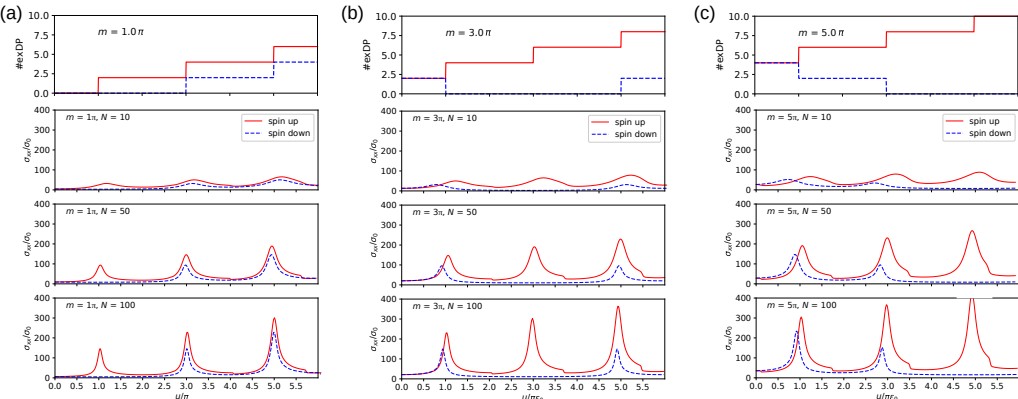

**Figure 7.** Conductance of spin-up (red line) and spin-down (dashed blue line) electron in *x* direction of finite zero-sum fields of PEP-PExP graphene superlattice at zero energy with variation of *u* and $N = 10, 50$ and 100 with (**a**) $m = 1.0\pi$, (**b**) $m = 1.0\pi$, and (**c**) $m = 5.0\pi$.

### 3.3. Electrical Controllable Spin-Dependent Band Structure

To explore more details of the influence of the exchange interaction potentials on the superlattice miniband that was correlated with the zero-average height of PEP [42], we

initialized $u_A = -u_B = u$ (or $\bar{u} = 0$), which were zero-averaged along the modulation direction and coupled with non-zero-averaged exchange fields, where $m_A \neq -m_B$ (or $\bar{m} \neq 0$). Consequently, we illustrate the model in Figure 8a.

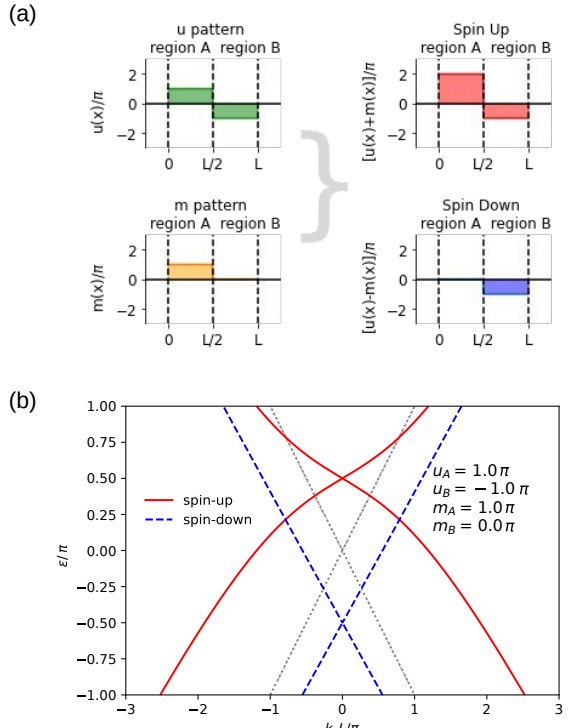

**Figure 8.** (**a**) Configuration of supercell profiles in PEP-PExP graphene superlattice, where $u_A = -u_B = 1\pi$ coupled with $m_A = 1\pi$ and $m_B = 0$. (**b**) Its corresponding miniband structure at $k_x = 0$.

As shown in Figure 8b, the numerical result revealed a miniband shifting upward for spin-up and downward for spin-down. To unveil this miniband splitting, analytically, we utilized the implicit function theorem on Equation (9) to locate its vertex points, which were the miniband touching points. In doing so, we set $\partial f(k_y, \varepsilon)/\partial k_y = \partial f(k_y, \varepsilon)/\partial \varepsilon = 0$ [43], where $f(k_y, \varepsilon)$ is a zero function of Equation (9) with $\kappa_x = 0$. This relation is satisfied only if $\cos(q_A w_A) = \cos(q_B w_B) = 1$ and $\sin(q_A w_A) = \sin(q_B w_B) = 0$, so we obtained

$$q_A w_A = \left( \sqrt{\frac{(\varepsilon - [u_A + m_A])^2}{\hbar v_0} - k_y^2} \right) w_A = m\pi, \tag{18}$$

and

$$q_B w_B = \left( \sqrt{\frac{(\varepsilon - [u_B + m_B])^2}{\hbar v_0} - k_y^2} \right) w_B = n\pi. \tag{19}$$

Here, we assume that the original Dirac point splitting occurs in the energy axis only so that we can consider Equation (18) with $(k_x, k_y) = (0, 0)$. Here, the first valid energy near the zero-energy point for this miniband is located at

$$E_{\mathbf{k}=0} = \frac{[u_A + m_A]w_A + [u_B + m_B]w_B}{w_A + w_B}. \tag{20}$$

With our preset, Equation (20) (with $u_A = -u_B = u$) becomes $(m_A + m_B)/2$. Accordingly, the spin-splitting behavior does not correlate with the electrostatic modu-

lation; instead, it relates to the exchange field modulation only corresponding to the miniband displacing.

Furthermore, we noticed that the anisotropic behavior of each spin miniband varied. The result of the effective period potential was different for each spin when $u = m$, as shown in Figure 8a. The modulation for each spin was enhanced for spin-up but suppressed for spin-down in region A. In this model, this phenomenon can be used to control the behavior of the miniband. To understand how to control this spin-dependent behavior, we tuned only the PEP profile by varying the magnitude of $u = -u_A = u_B$, while $m_A$ and $m_B$ remained fixed at $\pi$. As shown in Figure 9a, the periodic potential result for $u \geq m$ depends on the spin index of a charge carrier. In this case, the spin-up carrier was subjected to a modulated field that was stronger than that of the spin-down carrier. The spin-up minibands (see Figure 9b) strongly broadened on the $k_y$ axis. On the contrary, as illustrated in Figure 9c, the spin-down miniband was reshaped less. Extra band touching points can emerge in the miniband faster during spin-up than during spin-down, while the magnitude of $u$ increases. With this configuration, the extra Dirac point can emerge in spin-up but not in spin-down, resulting in highly spin-dependent behavior. Modulation of the effective periodic potential was lower for the condition $u \leq m$, as shown in Figure 9d. Spin-up was still periodically modulated, but spin-down was not, and the effective potential for each region varied marginally. As a result, the anisotropy of the miniband for the spin-down electron was nearly nullified, as shown in Figure 9f, while the anisotropy of the spin-up electron remained the same.

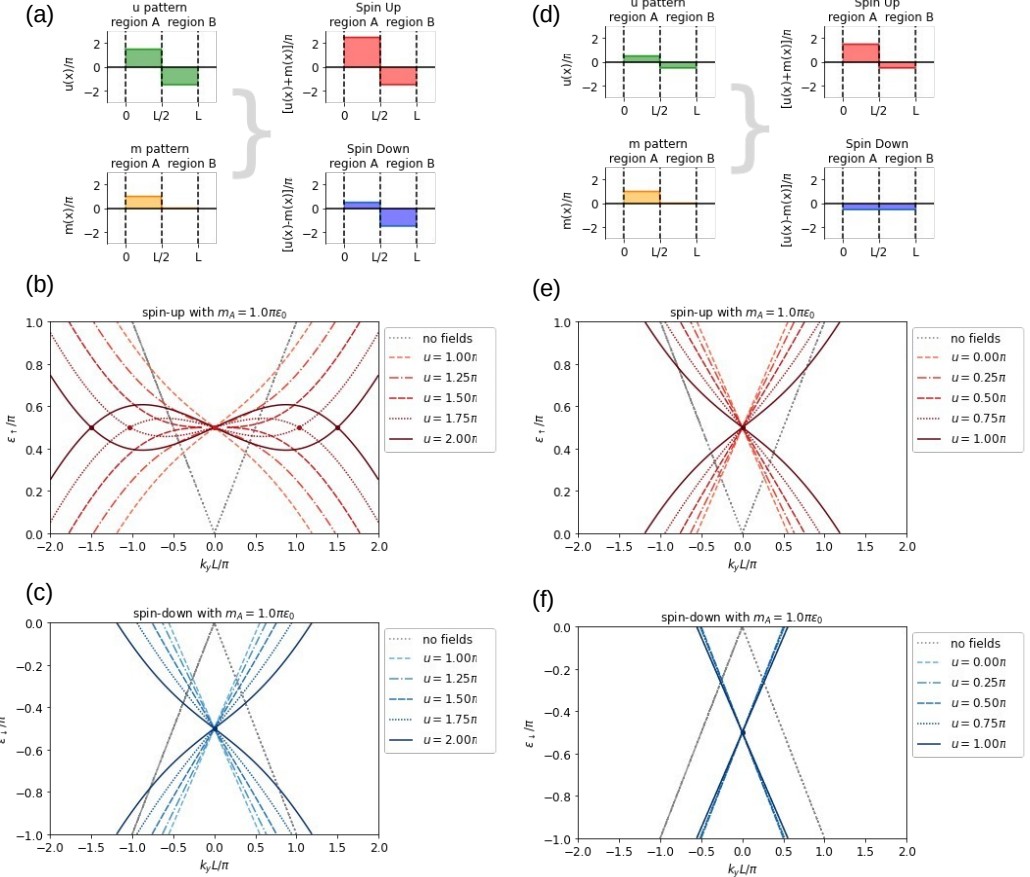

**Figure 9.** Supercell profiles and its corresponding miniband structure of the PEP-PExP graphene superlattice that fixed $m_A = 1\pi$ and $m_B = 0\pi$. (**a**) Pattern (1): superlattice profile in which $|u_{A,B}| = u \geq m_A$ ($u = 1$), and corresponding (**b**) spin-up and (**c**) spin-down minibands with varying $u$ such that $u \geq m_A$. (**d**) Pattern (2): superlattice profile where $|u_{A,B}| = u \leq m_A$ ($u = 1$) and corresponding (**e**) spin-up and (**f**) spin-down minibands with varying of $u$ such that $u \leq m_A$.

The anisotropic behavior of the spin-up miniband was enhanced as $u$ increased for these two conditions. The combination patterns between zero-averaged PEP and non-zero-averaged PExP are important in spin-dependent anisotropic minibands. As a result, we can tune the amplitude of the electrostatic modulation amplitude in the coupling pattern to control this spin-dependent behavior, because one spin band behaves as an anisotropic band, while the other spin does not.

To retrieve the real values of $U$ and $M$ in the discussion, we applied the unit of energy $\varepsilon_0 = \hbar v_0/L$ to restore $U$ and $M$. In doing so, $U = u\varepsilon_0 = u(\hbar v_0/L)$ and $M = m\varepsilon_0 = m(\hbar v_0/L)$. Notice that $U$ (and $M$) is inversely related to $L$, in which $U \propto 1/L$ (and $M \propto 1/L$). Consequently, if $U$ or $M$ is increased, to maintain the feasibility of the experiment result at a state of $u$ or $m$, the size of $L$ should be reduced. For example, if we set an appropriate value of $L$ at 10 nm and $u = m = 1.0\pi$, then $U = M = 1.0\pi\varepsilon_0 = 0.21$ eV.

## 4. Conclusions

In summary, we investigated the anisotropic spin-dependent behavior of the electronic properties in graphene with the alternating zero-averaged PEP/PExP superlattice using the transfer matrix method. Here, we introduced a PEP/PExP modulation pattern that is aligned as in-phase and out-of-phase with a zero-average field. This modulation model can provide a superlattice miniband of both spin-up and spin-down electrons whose original Dirac points are located at the same location in k-space. We investigated the influence of the combined field of PEP/PExP as an effective total field to control spin-dependent anisotropic behavior. As a result, for the in-phase modulation, the miniband of spin-up was highly deformed, while the spin-down one was not, and vice versa. Furthermore, the behavior of the spins can be manipulated by adjusting the strength of PEP and PExP. We also examined the anisotropic behavior of PExP and PEP that were zero-averaged along the modulation direction using the group velocity around zero energy. We found that the anisotropic velocity was spin-dependent, and the velocity of one spin index miniband was strongly suppressed by a stronger periodic modulation than the opposite one. For instance, in the in-phase modulation, at $u = m$ and $u \mod \pi = 0$ ($m \mod \pi = 0$), spin-up was strongly collimated, but the same effect on spin-down vanished. For the out-of-phase modulation, the behavior of spin-up and spin-down were switched. In addition, to extend our analysis on the above result, we also determined the location and number of minibands for each spin index. As a result, the spin dependence of the superlattice extra Dirac cone at zero energy caused additional Dirac points to emerge differently. We also clarified the spin-dependent anisotropic behavior on a finite superlattice by computing the zero-mode conductance of this model. Furthermore, we also found, upon controlling the spin-dependence of the alternating patterns of PEP/PExP, that PEP was zero-averaged, but PExP was not. We found that, with different total effective fields, it could enable spin-splitting features, including tuning the anisotropy of the miniband, as we introduced. This study sheds light on the possibility of building an electron spin-dependent collimator in a graphene-based nanodevice.

**Author Contributions:** Conceptualization, W.L.; methodology, P.S. and W.L.; validation, W.L.; formal analysis, P.S. and W.L.; investigation, P.S. and W.L.; writing—original draft preparation, P.S.; writing—review and editing, W.L.; visualization, P.S.; supervision, W.L.; funding acquisition, W.L. All authors have read and agreed to the published version of the manuscript.

**Funding:** This research was funded by Thailand Center of Excellence in Physics grant number (ThEP-61-PHY-KMUTT6).

**Data Availability Statement:** Data sharing is not applicable to this article.

**Acknowledgments:** P. Somroob gratefully acknowledges the financial support of the Science Achievement Scholarship of Thailand (SAST). Moreover, W. Liewrian thanks the Ministry of Higher Education, Science, Research and Innovation of Thailand for financially supporting this research with a grant under the Thailand Center of Excellence in Physics.

**Conflicts of Interest:** The authors declare no conflict of interest.

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
