# Peer review of "Electrical Manipulation of Spin-Dependent Anisotropy of a Dirac Cone in a Graphene Superlattice with Alternating Periodic Electrostatic and Exchange Fields"

_condensedmatter, doi:10.3390/condmat8010028_

Round 1

Reviewer 2 Report

The authors studied the electronic properties of graphene exposing to a mixed electrostatics and exchange field, and find the shifting of the spin-dependent shifting of the miniband. Moreover, the velocity anisotropy can be modulated by changing the potential profile. It is a interesting results, and I would recommend it to be published after some minor modifications.

Here are my comments.

1, It would be better if the authors could discuss more on the difference between their work and reference 17 which also reported the anisotropy of Dirac Cone in graphene superlattice.

2, To guide the experimental work, the authors should put in some real numbers. It can help the readers to evaluate the feasibility of such a proposal. 

Reviewer 3 Report

The paper  presents some well-known results on graphene in 1D periodic potential, studied by Peeters group more than 10 years ago.  Essentially,  each spin component propagates in the independent square potential that can be tuned.  The fact that the electrons with opposite spin orientations have very different properties under suggested potentials is obvious and there are much simpler methods to make a spin-filter having an exchange field at ones disposal.   The authors derive the dispersion and study it as a function of k_y and potential for k_x = 0.  Unfortunately, I have not found anything new in the results  and I leave it up to Editor if this poorly-written paper can be published after a major improvement of presentation and English. 

Round 2

Reviewer 3 Report

Thank you for your effort to improve the manuscript.  You must be aware that language editing service does not guarantee that the meaning is preserved (as, sometimes, the meaning is indeed unclear).  For example, already in the abstract you write:

"Spin-dependent collimation occurs when the modulation becomes zero— average  potentials with one spin can be moved freely, but the other one is highly collimated."    This is incomprehensible. 

"It  was shown that the peaks occur when the extra Dirac point emerges."

Peaks of what?  Conductance?

"Additionally, extra Dirac points may appear as  emerging zero modes [21] and additional Dirac points [22–28]" .  Extra  Dirac points appear as additional Dirac points????  Does this carry any meaning?

", we estimated the electronic behavior through the set n → ∞"  ,  probably, you  mean  "by setting  n-> inifinity" 

I have understood from the new clarifications that the novelty of this work is to say that all these effect studied before can be separately tuned for different spin orientations if one combines the exchange field and the external potential of comparable magnitudes.  Not a big deal, so leave it to the Editors if they want to publish it. 
